# Explaining the Correlates of the Multi-Theory Model (MTM) of Health Behavior Change in Visual (Structural) Colorectal Cancer Screening Examinations

**DOI:** 10.3390/ijerph22010098

**Published:** 2025-01-12

**Authors:** Asma T. Awan, Timothy J. Grigsby, Christopher Johansen, Chia-Liang Dai, Manoj Sharma

**Affiliations:** 1Department of Social and Behavioral Health, School of Public Health, University of Nevada, Las Vegas, NV 89119, USA; timothy.grigsby@unlv.edu (T.J.G.); christopher.johansen@unlv.edu (C.J.); manoj.sharma@unlv.edu (M.S.); 2Department of Teaching and Learning, College of Education, University of Nevada, Las Vegas, NV 89102, USA; chia-liang.dai@unlv.edu; 3Department of Internal Medicine, Kirk Kerkorian School of Medicine, University of Nevada, Las Vegas, NV 89102, USA

**Keywords:** colorectal cancer (CRC), multi-theory model (MTM), screening, colonoscopy

## Abstract

Colorectal cancer (CRC) ranks third in terms of global cancer prevalence and is the second most common cause of cancer-related mortality. Although CRC rates are decreasing in the United States, inequalities still exist despite the effectiveness of invasive screening methods, such as colonoscopy, flexible sigmoidoscopy, and computed tomography (CT) colonography in detecting colorectal cancer. Many current interventions promoting CRC screening do not utilize a modern theory-based approach, which has led to the low utilization of these screening methods. This cross-sectional study aims to address the lack of theory-based treatments for promoting visual CRC screening examinations by applying the multi-theory model (MTM) of health behavior change to explicate the health-related factors for individuals to seek visual colorectal cancer screening examinations for CRC screening. A 57-item validated questionnaire assessing MTM constructs and CRC screening was administered online. The survey questionnaire was administered to a sample of 640 adults from the United States. The participants were between the ages of 45 and 75 years. Hierarchical multiple regression was used to assess the relationship between MTM constructs with the initiation and sustenance of CRC screening behaviors. Out of the total participants in this nationwide sample, 71.4% (n = 457) reported that they had undergone a visual CRC screening examination. MTM subscales, specifically participatory dialogue, changes in the physical environment along with age, recommendation for CRC screening from a healthcare provider, and previous experience with colonoscopy, were found to be significant factors in predicting the initiation of visual CRC screening behavior. These factors accounted for 22% of the variation in initiation among this group (R^2^ = 0.222, F = 3.521, *p* < 0.001). The MTM can be a valuable framework for designing educational media, information media, social media platforms, and clinical interventions to promote visual colorectal cancer screening examinations.

## 1. Introduction

Colorectal cancer (CRC) is the third most prevalent cancer globally, comprising over 10% of all cancer cases, after breast and lung cancer [1]. CRC is the second leading cause of cancer-related fatalities worldwide, impacting the elderly population, with most instances occurring in adults aged 50 and above [1]. By 2030, the global incidence of all CRCs is projected to increase by 60% (2.2 million new cases) with mortality rates expected to rise by 22.6% [2,3,4]. In the United States (US), the overall annual, age-standardized CRC incidence rate has decreased by 46%, from 66.2 per 100,000 in 1985 to 35.7 per 100,000 persons in 2019, yet it is still on the rise in other countries [5,6]. In the US, this decline can be linked to changes in risk factors, including lower smoking rates, higher utilization of nonsteroidal anti-inflammatory medicines, adoption of CRC screening among adults aged 50 years and above, and expanded Medicare coverage [7,8,9,10].

While the uptake of colonoscopy among persons aged 50 years and older increased threefold from 20% in 2000 to 61% in 2021, the rate of uptake remained 20% for individuals in the age range of 45–49 years [5,11]. It is because of this reason that it is estimated that more than 1 in 3 adults ages 50 to 75 are not getting screened for CRC as recommended by the US Preventive Services Task Force (USPSTF) [12]. Further, less than 20% (fewer than 4 million out of the eligible 19 million) were found compliant with the recommended screening in 2021, and the rate of screening decreased from 19.5% in 2019 to 17.8% in 2021 in the general population [13]. The proportion of American adults who never have had CRC screening declined from 27.4% in 2012 to 21.6% in 2020, reflecting a reduction of 5.8 percent [13]. It is also reproted that this decrease corresponds to about 3.9 million fewer individuals checked for CRC in 2012 compared to 2020 [14]. The most common screening tests received by almost 68% of adults aged 50–75 years from 2000–2021, were visual structural examinations [15].

Evidence indicates that some screening examinations for CRC effectively detect cancer in its early stages and might reduce mortality rates associated with the condition. The USPSTF has recommended that adults aged 45 to 75 years be screened for CRC, and individuals should make an informed, personalized decision on whether to undergo screening between the ages of 76 and 85 [12]. CRC tests are broadly categorized as follows [16]:

### 1.1. Stool Tests

Guaiac-based fecal occult blood test (gFOBT),Fecal immunochemical test (FIT), andFecal Immunochemical Test and DNA Test (FIT-DNA test).

### 1.2. Visual Tests

Flexible sigmoidoscopy,Colonoscopy, andComputed tomography (CT) colonography or virtual colonoscopy)

Individual behavioral characteristics play a vital role in determining CRC screening uptake behavior. In adults aged 55 years and older who never had any CRC screening, health education increased general awareness, reduced psychosocial barriers, and mediated risk perception [17]. A plethora of behavioral interventions exist to increase the uptake of screening colonoscopy–the most widely adopted interventions targeted patient navigation behavior while increasing the uptake and completion of colonoscopy [18]. Behavioral therapies, as explained by Yakoubovitch and colleagues, enhance the likelihood of patients successfully adopting screening colonoscopies and its implementation in clinical practice could specifically, identify barriers to CRC screening, including costs required to initiate and sustain these effective interventions [18]. The sceening costs vary from USD 1700 to USD 2700 for visual procedures, and from USD 10 to USD 800 for the non-visual CRC screening prpcedures [19]. Importantly, these tests might not captute the true cost-effectiveness, if the the patients are reffeed for subsequent surgical or invasive procedures. Therefore, to implement colonoscopy as the screening modality, it is necessary to inform patients of the risks and advantages of the procedure [20]. This consensus should center on potential damage, patient-informed choice and decision, socioeconomic concerns, and the enhancement of colonoscopy quality, as well as the effectiveness of the procedure in preventing CRC. Research is scarce regarding the role of behavioral theories in elucidating the factors influencing attitudes and actions related to CRC screening [21]. Some theories that have been used are the health belief model, the protection motivation theory, and the theory of planned behavior [22,23,24]. Theory-based predictors are also helpful in designing and implementing educational interventions in this area but do not explicitly examine the behaviors for visual examination for CRC screening.

Choosing a suitable theory is crucial in CRC research. The novel, fourth-generation Multi-Theory Model (MTM) of Health Behavior Change may be especially useful for CRC research as it can be used to explain correlates of uptake and sustained screening efforts over the life course [25]. The MTM theory posits that behavior change involves two main stages, initiation and sustenance or maintenance (Figure 1). During the initiation process, an individual must be persuaded that the benefits of a particular action are greater than its drawbacks (through participatory dialogue). Additionally, individuals must possess a strong belief in their ability to carry out the desired behavior and receive support from the surrounding physical environment. To sustain a behavior change, an individual needs to effectively convert their emotions into objectives, consistently work towards practice for change, and receive support from their social surroundings. The theory has been applied effectively to predict various health behaviors, such as mammography screening, HPV vaccination, cervical cancer screening, and stool-based CRC screening [26,27,28,29,30].

In the present study, the health-seeking behavior for visual examination modalities for CRC screening tests (colonoscopy, flexible sigmoidoscopy, CT colonography) was analyzed among a nationally representative sample of people between the ages 45 to 75 years old. As of now, and to our knowledge, this is the first exploratory study using MTM for visual examination techniques for CRC screening in US adults aged 45–75 years.

## 2. Materials and Methods

### 2.1. Participants and Procedure

A US-wide cross-sectional survey was administered between January and March 2023 to eligible US adults aged 45–75 years using a Qualtrics (Provo, UT) web-based panel [31]. Eligibility criteria were age between 45 and 75 years, having undergone any CRC screening test, any conditions related to colon disease, and current personal and family history of CRC [30]. The sample size for the present analysis was determined by utilizing the parameters of a confidence level of 95%, a margin of error of 5%, and a population proportion of 33% derived from the data provided by the Centers for Disease Control and Prevention in 2018 [32]. These data showed that 67% of adults in the United States between the ages of 50 and 75 met the recommended guidelines for CRC screening [32]. A minimum sample size of 340 was required. An attrition rate of 20% was considered for non-response. Finally, a total of 408 participants were considered for sampling.

A total of 1060 individuals were invited for the survey, but 139 (13.11%) left the informed consent blank, 42 (3.96%) did not provide consent, 11 did not meet the age criteria, 55 (5.18%) had missing age information, 39 (3.67%) did not complete the survey, and 134 (12.64%) participants had missing information about CRC cancer screening (Figure 2). The final sample (n = 640) had less than 5% of the missing values on variables of interest which were the demographics, personal and family history of Crohn’s disease, history of non-stool-based CRC screening test (flexible sigmoidoscopy, colonoscopy, computed tomography (CT) colonography or virtual colonoscopy), and the likelihood of getting a visual screening test for CRC.

### 2.2. Measures

#### 2.2.1. Multi-Theory Model of Behavior Change

The study utilized an MTM-based instrument in which the initiation phase of the theory was synergized with the theoretical construct of changes in the social environment from the sustenance model assuming the unpredictable duration of the behavior change. The survey tool describes a 57-item questionnaire devised by the MTM theory originator and primed through a two-stage expert panel review. During the expert review procedure, seven specialists were asked to evaluate the questionnaire’s content and face validity. Two of the panel members were experienced in the field of internal medicine and CRCs; two panel members had skills in target population screening and preventive care; one expert was outside of public health; and two members were the content experts from health behavior research, instrument development, and public health theories, models, and frameworks (TMFs). The experts were asked to evaluate the questionnaire items in terms of their clarity, readability, and relevance. The language of the items was slightly altered per the feedback provided by the experts. All the experts endorsed the content and face validity of each of the MTM subscales for the acceptability of the survey. The instrument achieved a reading ease score of 62.4 and a Flesch–Kincaid Grade Level of 7.7, indicating its readability level. A “good readability score” on the Flesch Reading Ease scale is typically regarded as being approximately 60 or above, which aligns with an 8th-grade reading level, thereby ensuring that the content is accessible to a broad audience. In terms of the Flesch–Kincaid Grade Level, a score ranging from 6 to 9 is deemed satisfactory, reflecting an average reading level appropriate for middle school students.

#### 2.2.2. Questionnaire

The questionnaire begins with 13 screening questions about the age and the respondents’ experience undergoing any CRC screening test during the past one to five years. This is followed by 21 questions directed toward gathering clinical, medical, and family history of CRC and demographic information, including age, gender, race or ethnicity, religion, geographical location, marital status, and employment status. The subsequent, final 21 items assess the components of the Multi-theory Model (MTM) for both its initiation and maintenance phases for CRC screening as shown in Figure 1. There are two main phases of the MTM, “initiation”, and “sustenance”. For the current study, only the “initiation” phase was operationalized because of the sporadic nature of these tests. The modified phase of “initiation” had four constructs, *participatory dialogue, behavioral confidence, changes in the physical environment*, and *changes in the social environment*—which was derived from the sustenance phase. The first construct of the *participatory dialogue* is calculated from the *perceived advantages* and *perceived disadvantages* (participatory dialogue score = perceived advantages minus perceived disadvantages). There were 5 question items for the *perceived advantages* and *perceived disadvantages*, respectively, and mapped on a 5-point Likert scale ranging from 0 “Never” to 4 “Very often” with the range lying within 0–20 units. The second construct was *behavioral confidence* which was calculated on the certainty scale with the range of 0–20 units on a Likert scale as well. The third construct in “initiation” is the *changes in the physical environment*, which refers to the physical capacities needed to start a specific behavior. This was also measured on a 5-point Likert scale for three of its items. The last construct of *changes in the social environment* is based on the impact of social norms and effects on initiating a behavior and is again measured on the 5-point Likert scale for its three items. Each construct of MTM was assessed as a separate predictor to initiate the behavior.

### 2.3. Outcome Variable

The outcome variable of interest was derived from the modeling initiation question, “How likely is it that you will get a colorectal cancer screening when it is due”? The 5-point Likert Scale measured: not at all likely (0), somewhat likely (1), moderately likely (2), very likely (3), and completely likely (4). The time frame was attributed to the past 5–10 years depending on the type of visual examination for CRC screening, i.e., colonoscopy every 10 years, CT colonography (virtual colonoscopy) every 5 years, and flexible sigmoidoscopy (FSIG) every 5 years.

### 2.4. Covariates

Demographic variables used were for age in years (continuous variable), gender (male, female, other, preferred not to answer), race (AAPI, Black, White, other), ethnicity (Hispanic, Non-Hispanic), marital status (married, divorced/separated, widowed, single/never married, other), education (less than a high school, high school diploma or GED, some college but no degree, college degree, graduate-level degree, other), presence of health insurance (yes/no), employment status (employed or self-employed, not working, unable to work), religion (non-Christianity, Christianity), and median income (<USD 25,000; USD 25,000–USD 50,000; USD 50,001–USD 75,000; USD 75,001–USD 100,000; USD 100,001–USD 125,000; USD 125,001–USD 150,000; >USD 150,000).

Other variables included a family history of CRC (yes/no) and personal history of Crohn’s disease (yes/no), and a history of undergoing any visual, non-stool-based CRC screening test (flexible sigmoidoscopy, colonoscopy, computed tomography (CT) colonography or virtual colonoscopy), measured as yes/no with no as the reference category.

### 2.5. Ethical Considerations

The Institutional Review Board at the University of Nevada, Las Vegas approved study procedures. No personal identifiers were collected or gathered to guarantee the anonymity of the participants.

### 2.6. Data Analysis Plan

The data were checked for assumptions of independence of observations and tested for linearity (scatterplot with a line of best fit). Other assumptions were checked for homoscedasticity, multicollinearity, and normality (i.e., histogram, P–P plot, and Q–Q plot) and were examined for this study before selecting the model. For initiation modeling, a hierarchical multiple regression (HMR) analysis was utilized for those participants who would likely undertake the CRC screening in the form of visual CRC screening examinations (Colonoscopy, CT Colonography, and Flexible Sigmoidoscopy). For HMR predictor variables are added to the initial model in a predetermined manner based on theory which are important to see how much variance in the dependent variable is explained by each new variable added. For our analysises, five models were run for HMR. Model 1 included covariates, model 2 included model 1 + participatory dialogue, model 3 included model 2 + behavioral confidence, model 4 included model 3 + changes in the physical environment, and the final model 5 included model 4 + changes in the social environment. The variables’ selection for the HMR model was synthesized and derived from the theoretical concepts of MTM and statistical underpinnings. All analyses were conducted on IBM SPSS (version 29.0) [33], with a level of significance set at 5%.

## 3. Results

As shown in Table 1, the mean age was 58 years (SD = 8.96) and 71.4% (N = 457) reported having undertaken CRC screening in the form of visual CRC screening examinations (Colonoscopy, CT Colonography, Flexible Sigmoidoscopy). Most participants identified as non-Hispanic White (51.7%), Christians (67.0%), and were married (47.3%). The sample was predominantly female (57.6%), non-Hispanic (80.2%), and reported having health insurance (88.5%). A higher number of participants in the sample had some college degree (32%). More than half of the sample reported previously undergoing a colonoscopy (52.1%) as the modality of test for visual CRC screening examinations.

### Correlation of MTM Constructs

As shown in Table 2, the construct of “participatory dialogue” demonstrated a positive and moderate correlation with the constructs of “behavioral confidence”, “changes in the physical environment”, and “changes in the social environment” (r = 0.51, r = 0.45, and r = 0.42, respectively; *p* < 0.001 for all). In addition, the construct of “behavioral confidence” also demonstrated a strong positive correlation with the “changes in the physical environment” and “changes in the social environment” (r = 0.75 and r = 0.68, respectively; *p* < 0.001 for all). The “changes in the physical environment” were strongly and directly correlated with the “changes in the social environment” (r = 0.77; *p* < 0.001).

Model 1 (covariates only) of the hierarchical regression model (Table 3) explained 13% of the variance in CRC screening. Age was negatively associated with screening *p* < 0.001, but other covariates were not. Model 2 (model 1 + participatory dialogue) explained an additional 7% of the variance in CRC screening. Controlling for covariates, there was a significant positive association between participatory dialogue and CRC screening. Model 3 (model 2 + behavioral confidence) did not indicate any change in explaining the variance in CRC screening (ΔR^2^ = 0.00). Model 4 (model 3 + changes in the physical environment) explained an additional 11% variance in CRC screening. Model 5 (model 4 + changes in the social environment) explained an additional 4% of the variance in CRC screening but the relationship between changes in the social environment and CRC screening was non-significant. Overall, the MTM subscales of “participatory dialogue”, and “changes in the physical environment” were the only theory-based significant predictors of initiating screening behavior in the final model (model 5). Specifically, for every one-unit increase in participatory dialogue, there was a −0.022 change in CRC screening (*p* < 0.001). Also, for every one-unit increase in changes in the physical environment, there was a 0.028 increase in CRC screening behavior (*p* < 0.001).

Among covariates, recommendations for CRC screening from a healthcare provider had a regression coefficient of 0.136, implying that the predicted value of CRC screening increased modestly for a positive recommendation as opposed to no recommendation (β = 0.160, *p* < 0.05). Previous experience with colonoscopy had a regression coefficient of −0.116, implying that the predicted value of CRC screening decreased for those who had previous experience with colonoscopy (β = −0.127, *p* < 0.001).

## 4. Discussion

This study aimed to identify theory-based constructs associated with the utilization of visual CRC screening examinations (i.e., flexible sigmoidoscopy, colonoscopy, computed tomography (CT) colonography, or virtual colonoscopy) for CRC among persons aged 45–75 years, employing the modified initiation theoretical paradigm of the multi-theory model (MTM) of health behavior change. The overarching conclusion of this exploratory study was that the construct of changes in the physical environment along with age and recommendation for CRC screening from a healthcare provider were significant and positively associated with visual CRC screening, while the construct of participatory dialogue and previous experience with colonoscopy were significant and negatively associated with visual CRC screening. These factors, adjusting for other covariates, accounted for 22% of the variation which is an encouraging outcome for health behavior research [34].

A conducive physical environment is essential for an older adult to seek visual CRC screening. This includes overcoming barriers in CRC screening visual procedures which previous studies have identified such as facilitators and barriers for colonoscopy as a modality for CRC screening tests [35]. Our study has implied that environmental factors count significantly toward the adoption and follow-up of any CRC screening test [36]. Understanding of CRC and screening, as well as the perspectives of patients and providers, or the systemic barriers to screening are some of the factors which may also determine how individuals take up and adopt the visual CRC screening tests. Although these studies are valuable for identifying groups with low participation rates, they mostly concentrate on demographic, clinical, and program-related characteristics rather than the crucial reasons why patients refuse the test offer [37]. To achieve this objective, as explained by Dalton [37], a diverse array of qualitative studies should be conducted.

Our study is the first to quantitatively apply and understand the MTM theoretical framework for utilizing any visual CRCs screening test with factors related to changes in the physical environment. The construct of changes in the physical environment also played a crucial role in the initiation of behaviors in other studies of MTM [27,29,38,39,40,41]. The features of the physical environment have been verified as strong variables, associated with factors or activities directly or indirectly derived from the physical environment. This environmental enrichment has been deemed necessary from previous work done on the multi-theory model (MTM) of health behavior change. This is an important addition to a previous study done with stool-based tests, which did not find any association between changes in the physical environment and the intent to undertake stool-based CRC screening [30]. The construct of participatory dialogue had a negative association—which was unexpected.

One possible explanation for the construct of participatory dialogue that had a negative association is that it accentuates the role—that the disadvantages of visual CRC screening play in preventing people from seeking these tests and highlights the role of educational interventions. Likewise, previous exposure to visual CRC screening was also found to be a significant hindrance in the intention of people to undertake visual CRC screening [42,43]. This refers to the fact that many people avoid these procedures. Here stool-based tests particularly Cologuard [44] or multi-targeted stool DNA test (mt-sDNA) can play an important conduit in routing visual screening to selective colonoscopy because of the noninvasiveness and potential to increase participation and clinical outcomes [45,46,47]. Behavioral confidence and changes in the social environment were not significant constructs in our study. It could be because visual CRC screening procedures are dependent on the healthcare provider, while family and friends and one’s confidence may not be that crucial. Research on the impact of family involvement in psychosocial interventional programs on the outcomes of colorectal patients is highly recommended because psychosocial therapies that include family members have been shown to have positive effects [48]. This underscores the role that the disadvantages of visual CRC screening play in preventing people from seeking these tests and highlights the role of educational interventions

Furthermore, a recommendation from a healthcare provider also played a significant role in this study. This is concordant with previous studies in which healthcare providers were considered to provide patients with accurate suggestions based on scientific information on sources and adhere to evidence-based guidelines for cancer screening [49]. The healthcare providers’ role is particularly important in the context of CRC, especially their motivation towards effective screening programs can significantly reduce the incidence of this neoplasm [50]. It cannot be overemphasized that healthcare providers, especially in primary care settings, need to devote time to counsel their patients on the merits of visual CRC screening at prescribed intervals.

Aging is also positively associated with visual CRC screening and can be explained on the basis that screening colonoscopy may have provided a better advantage in avoiding CRC in individuals aged 70 to 74 years, as compared to a lesser advantage in individuals older than 75 years [51]. This aligns with a targeted approach to screening for older patients with significant life expectancies and avoiding screening older individuals with limited life expectancies [52]. CRC screening is determined by the role of increasing age with increasing genetic expression and the impact of age-related biomarkers on the prognosis of CRC patients [53]. The likelihood of experiencing negative experiences was minimal, although it was higher among those of advanced age. This aligns with the need for optimal, diagnostic yield for colonoscopy and the development of highly sensitive, population-based CRC screening systems [54].

Recent breakthroughs and advancements in potential blood tests also offer some promise to restricting visual CRC screening in selected cases [55]. Also, the role of Artificial Intelligence (AI) in assisting patients undergoing preparation for visual CRC screening is being applied which will make these tests easier. AI technologies and computer-aided detection (CADe) tools can assess bowel preparation [56], detect suspicious lesions that humans may overlook [57], and quantify polyp detection and characterization with real-time audio-visual alerts [58]. Recent adoptions of AI techniques can grade the fold examination quality (FEQ) of each colonoscopy, after which the recorded colonoscopies are evaluated by the AI system [59]. The system assessed the correlation of FEQ and evaluated expert scoring, historical adenoma detection rate (ADR), and withdrawal time of each endoscopist directly related to the diagnosis of colon cancer. The AI system’s evaluations of the FEQ of each endoscopist were significantly correlated with expert scores, historical ADR, and withdrawal time [60].

### 4.1. Implications for Practice

Most studies on MTM have reported factors related to changes in the physical environment about the availability of resources, relevant literature, equipment, types of logistics, distances to and from the provider facilities, and means of transportation. In the current study, it can be related to the availability of resources and transportation to facilities with services providing visual CRC screening tests [48]. This situation may present quandaries and difficulties during the follow-up with primary care providers or practitioners after a screening stool-based exam and then getting a colonoscopy [61,62], which is the gold standard for diagnosing CRC [63]. To assist these individuals, it is essential to comprehend the underlying reasons behind their decision to reject colonoscopy, especially alleviating the barriers in the physical environment and making the environment conducive to CRC screening services [64,65]. Furthermore, healthcare practitioners must prioritize personalized recommendations for CRC screening. In addition to these, changes in the physical environment, educational programs led by healthcare providers must be the cornerstone of practice.

### 4.2. Recommendations for Research

To further our understanding of how altering environments might influence behavior, it is crucial to conduct well-designed field studies that accurately measure the impact of these changes. Additionally, clinical research should be conducted to investigate the mechanisms underlying these interventions and to establish a theoretical framework for understanding them [66]. The mean MTM scores of the group indicated that they did not achieve the highest potential scores, indicating a need for improvement. One way to achieve this is to implement educational interventions that aim to enhance performance in MTM components. In addition, the mean scores of all MTM initiation constructs showed a stronger directionality for not undergoing colonoscopy. Furthermore, these results provide additional evidence to corroborate the inferential findings using a qualitative and/or mixed-methods approach. They strongly suggest that interventions utilizing MTM as a behavior change model can effectively encourage undergoing a colonoscopy as the screening and diagnostic test.

This study is the first to utilize the fourth-generation framework of MTM to conceptualize and synthesize data on the factors associated with visual CRC screening for individuals between the ages of 45 and 75 years. The study successfully elucidated a significant portion of the variation in the initiation of visual CRC screening examinations within a specific demographic.

### 4.3. Strengths and Limitations

The study yielded findings that can be used to develop interventions aimed at encouraging the use of visual CRC screening examinations, especially colonoscopy, as the gold diagnostic standard test for CRC screening, even after stool-based CRC screening tests for the specific target population. Nevertheless, this study had several limitations. First, the cross-sectional study design yielded prompt outcomes but hindered the ability to establish causal relationships due to the absence of established temporality. Second, the analysis was based on self-reports, which are susceptible to various biases. Third, the study focused primarily on identifying attitudes that could have the objectivities of the researcher evaluations, leaving no alternative methods to achieve this goal. Fourth, data were not collected for partial replies, which affected the analysis of the characteristics of the non-responders. This analysis is crucial for understanding the reasons for non-response bias. On the other hand, there has been successful confirmation for the MTM-based instrument for face, content, and construct validity, as well as its reliability in terms of internal consistency and test-retest reliability.

## 5. Conclusions

This study resonated with the constructs of the modified framework of the multi-theory model (MTM) of health behavior change to explain the correlates of visual CRC screening examinations and initiation of screening tests for CRC among individuals aged 45–75 years. Particularly, the constructs of participatory dialogue and changes in the physical environment along with age, recommendations from healthcare providers, and previous experience with visual CRC screening were found to be significantly predictive of the decision to initiate any visual CRC screening examination as a screening test. Before engaging in intervention research, researchers should undertake a mixed-methods approach and reiterate the findings using the MTM.

## Figures and Tables

**Figure 1 ijerph-22-00098-f001:**
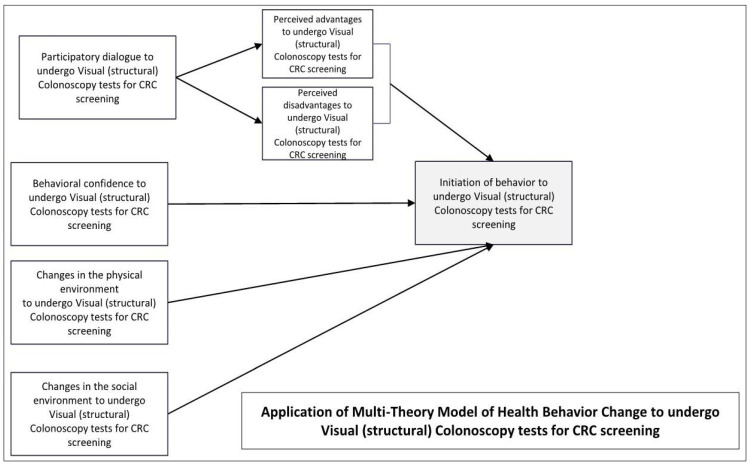
Constructs of multi-theory model of health behavior change (MTM) for visual CRC screening behavior.

**Figure 2 ijerph-22-00098-f002:**
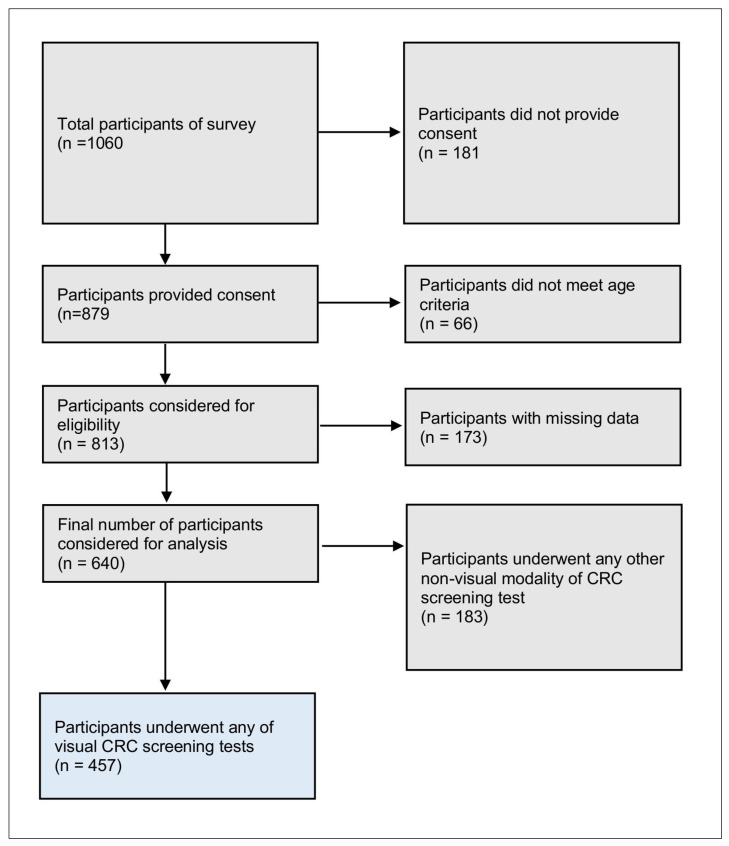
Flow diagram for selection of study participants who have taken any of the visual CRC screening tests.

**Table 1 ijerph-22-00098-t001:** Sociodemographic characteristics of the study groups (N = 457).

Variables	Categories	Overall Sample n (%)
Age in years (Mean ± SD)		58.26 ± 8.96
Gender	Male	271 (42.4)
	Female	368 (57.6)
Race	Black	74 (11.6)
	White	331 (51.7)
	AAPI	23 (3.6)
	Other	36 (5.6)
Ethnicity	Hispanic	86 (13.4)
	Non-Hispanic	513 (80.2)
Marital status	Married	303 (47.3)
	Divorced/Separated	123 (19.2)
	Widowed	42 (6.5)
	Single, never married	103 (16.1)
	Other	33 (5.2)
Education	Less than a high school	17 (2.6)
	High school diploma or GED	154 (24.0)
	Some college but no degree	161 (25.2)
	College degree	205 (32.0)
	Graduate level degree	61 (9.5)
	Other	7 (1.2)
Health insurance	Yes	567 (88.5)
	No	37 (5.8)
Region	Rural	176 (27.5)
	Urban	152 (23.7)
	Suburban	277 (43.2)
Employment Status	Employed or self employed	279 (43.5)
	Not working (e.g., out of work, homemaker, retired)	260 (40.6)
	Unable to work	66 (10.3)
Religion	Christian	429 (67.0)
	Non-Christian	176 (27.5)
Median income	<USD 25,000	143 (22.3)
	USD 25,000–USD 50,000	162 (25.3)
	USD 50,001–USD 75,000	111 (17.3)
	USD 75,001–USD 100,000	57 (8.9)
	USD 100,001–USD 125,000	40 (6.2)
	USD 125,001–USD 150,000	33 (5.2)
	>USD 150,000	37 (5.7)
Underwent any non-stool-based test	Colonoscopy	317 (52.1)
	CT Colonography	84 (13.8)
	FSIG	56 (9.2)

**Table 2 ijerph-22-00098-t002:** Pearson correlations between multi-theory model (MTM) constructs used in the study (N = 457).

MTM Construct	1	2	3	4
Participatory Dialogue	--			
2. Behavioral Confidence	0.510 ** [0.448, 0.567]	--		
3. Changes in the Physical Environment	0.447 ** [0.380, 0.508]	0.745 ** [0.707, 0.778]	--	
4. Changes in the Social Environment	0.417 ** [0.349, 0.481]	0.676 ** [0.631, 0.718]	0.765 ** [0.729, 0.796]	--

** *p* < 0.001. Numbers in brackets represent the 95% confidence interval for estimated correlation coefficients.

**Table 3 ijerph-22-00098-t003:** Hierarchical multiple regression to investigate predictors among those who have had any form of visual colorectal cancer screening.

Variables	Model 1		Model 2		Model 3		Model 4		Model 5	
	B	β	B	β	B	β	B	β	B	β
Constant	0.324		0.195		0.187		0.181		0.171	
Age	0.000 **	−0.006	0.002 *	0.042	0.002 *	0.042	0.002 *	0.039	0.002 *	0.044
Gender: Male (Ref: Female)	0.025	0.028	0.037	0.040	0.037	0.041	0.040	0.044	0.038	0.042
Race: White (Ref: Black)	0.026	0.044	0.031	0.051	0.031	0.051	0.028	0.047	0.027	0.044
Ethnicity: non-Hispanic (Ref: Hispanic)	0.106 *	0.054	0.074 *	0.038	0.074 *	0.038	0.081 *	0.041	0.080	0.040
Region: Urban (Ref: Rural)	−0.038	−0.072	−0.030	−0.056	−0.030	−0.056	−0.032	−0.060	−0.035	−0.066
Marital Status: Others (Ref: Married)	0.030	0.088	0.023	0.067	0.022	0.066	0.020	0.060	0.021	0.062
Education	0.002	0.006	0.015	0.035	0.014	0.035	0.010	0.024	0.009	0.021
Health insurance: Yes (Ref: No)	−0.078	−0.042	−0.062	−0.033	−0.064	−0.034	−0.091	−0.049	−0.078	−0.042
Employment status: Not Working (Ref: All Other Types)	−0.070	−0.102	−0.059	−0.086	−0.059	−0.086	−0.063	−0.092	−0.066	−0.096
Religion	−0.013	−0.013	−0.020	−0.019	−0.019	−0.019	−0.025	−0.025	−0.025	−0.024
Income	0.015	0.059	0.014	0.057	0.014	0.055	0.011	0.044	0.013	0.054
Personal history of colorectal cancer	0.030	0.088	0.033	0.011	0.031	0.010	0.037	0.012	0.038	0.012
Family history of colorectal cancer	0.030	0.088	−0.079	−0.057	−0.080	−0.058	−0.075	−0.054	−0.073	−0.053
History of IBD	0.030	0.088	−0.066	−0.043	−0.067	−0.043	−0.067	−0.043	−0.063	−0.041
History of Hereditary CRCs	0.030 *	0.088	0.254	0.098	0.254	0.098	0.261	0.101	0.242	0.093
History of Abdominal Radiation	0.030	0.088	0.129	0.066	0.132	0.067	0.111	0.057	0.107	0.055
Recommended CRC Screening	0.108 *	0.128	0.139 *	0.165	0.140 *	0.165	0.137 *	0.162	0.136 *	0.160
Colonoscopy	−0.191 **	−0.209	−0.104	−0.114	−0.105	−0.115	−0.117 *	−0.129	−0.116 **	−0.127
Virtual Colonoscopy	0.014	0.011	0.017	0.013	0.017	0.013	0.016	0.013	0.023	0.018
FSIG	0.062	0.043	0.019	0.013	0.018	0.013	0.028	0.020	0.035	0.024
Participatory Dialogue			−0.021 **	−0.310	−0.021 **	0.051	−0.022 **	−0.331	−0.022 **	−0.325
Behavioral Confidence					0.001	0.038	−0.008	−0.095	−0.006	−0.074
Changes in the Physical Environment							0.022 *	0.166	0.028 **	0.213
Changes in the Social Environment									−0.012	−0.097
R^2^	0.133		0.208		0.208		0.218		0.222	
F	2.285		3.721		3.541		3.597		3.521	
ΔR^2^	0.133		0.075		0.000		0.011		0.004	

IBD = irritable bowel disease, CRC = colorectal cancer, FSIG = flexible sigmoidoscopy; * *p*-value < 0.05; ** *p*-value < 0.001; adjusted R^2^ of model 5 = 0.159.

## Data Availability

The data presented in this study are available on request from the corresponding author. The data are not publicly available due to ethical reasons.

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
