# Peer review of "Explaining the Correlates of the Multi-Theory Model (MTM) of Health Behavior Change in Visual (Structural) Colorectal Cancer Screening Examinations"

_ijerph, 2025, doi:10.3390/ijerph22010098_

Round 1
Reviewer 1 Report
Comments and Suggestions for Authors
It’s a great topic to use the Multi-theory model to explain cancer screening behavior. However, there are a few questions regarding the study design.
1) Could you please provide more details on sampling strategy? How were 1060 participants invited to complete the survey (e.g., via mail or email) ? Can the sampled population be considered representative of the general US population?
2) I reviewed your reference 29, “Using the Multi-Theory Model (MTM) of Health Behavior Change to Explain the Seeking of Stool-Based Tests for Colorectal Cancer Screening”. In this previously published article, a sample of 389 participants was described as “not undergo stool-based CRC screening”, which I assume that 389 participants who either underwent visual CRC screening or did not undergo any screening test at all. However, in your manuscript, Table 1 presented the exact same demographic characteristics, yet 457 of them were reported as having visual CRC screening. Could you please clarify why these numbers appear to conflict?
Without clarification of sample size in current manuscript, the validity of the subsequent data analysis remain uncertain.
Author Response
Please see attached.
Best regards,

Reviewer 2 Report
Comments and Suggestions for Authors
Please see review attached.

Reviewer 3 Report
Comments and Suggestions for Authors
The authors of this article emphasized the effectiveness of the Multi-Theory Model (MTM) of health behavior change in identifying factors influencing colorectal cancer (CRC) screening behaviors, underscoring the significance of participatory dialogue and healthcare provider recommendations in predicting the initiation of visual CRC screening.
Please find my comments below:
1. I would suggest listing the stools tests and visual tests, instead of including them as examples in a list (lines 70 and 71).
2. The authors mentioned that “he USPSTF has recommended that adults aged 45 to 75 years be screened for CRC”, I would expand on this point to explain the frequency of the screening test (e.g., every two years using a FIT) and emphasize that this group is the target population for CRC screening.
3. There is a minor spacing issue (line 79) “explained byYakoubovitch and colleagues”.
4. The in-text citations for line 89 should be all together [21, 23, 23].
5. In the background, it would be interesting to include more information about the cost of stool-based versus visual CRC screening, as that could be barrier to individuals obtaining visual CRC screening examinations.
6. In the Participants and Procedures section, I would suggest providing more information about the variables of interest (line 133).
7. It would be interesting to include a flow-chart/visualization for the study sample, displaying the total number of participants invited, those who were excluded (E.g., 139 who left the informed consent blank), and the final number of study participants (640 participants). This would be similar to a PRISMA flowchart.
8. Under the Measures section, I would suggest separating the list with semi-colons instead of commas (lines 140-144),
9. When describing the readability score, I would suggest providing more context as to what is considered a “good readability score” for the reading ease score and Flesch-Kincaid Grade Level.
10. Under the questionnaire, I would suggest providing information as to how consent was obtained from participants. Alternatively, this could be included under ethical considerations.
11. There is a small error under line 154, where “twenty-one” is spelled out instead of written as numerals. Please pay close attention to how numbers are written out, as the general rule is to spell out numbers that are nine and under.
12. Line 169, there is a minor spacing error “Likert scale as well.”
13. The tables (tables 1-3) can be included under an appendix.
14. There should be an in-text citation for Figure 2, as it’s included in the body of the results, but it is not referenced within the paper.
15. Lines 327-330, could you elaborate on what is considered a conducive physical environmental and environmental factor you’re referring to.
16. This sentence “Other factors may also determine how individuals take up and adopt the visual CRC screenings.” Seems very vague, I would suggest revising and include specific factors from the results and/or compare factors noted in other studies (line 332-333).
17. I would be suggesting revising lines 340-345 to improve the clarify, as the various topics of the studies do not need to be explained, they can be generally stated “the construct of changes in the physical environment also played a role in other studies…”
18. There is a minor error where “CRC” is repeated twice (line 417).
Round 2
Reviewer 1 Report
Comments and Suggestions for Authors
Thanks. The edited version answered my questions.